# Implementation Activities in Smoke-Free Public Housing: The Massachusetts Experience

**DOI:** 10.3390/ijerph20010078

**Published:** 2022-12-21

**Authors:** Boram Lee, Vicki Fung, David Cheng, Jonathan P. Winickoff, Nancy A. Rigotti, Radhika Shah, Claire McGlave, Sydney Goldberg, Glory Song, Jacqueline Doane, Melody Kingsley, Patricia Henley, Sanouri Ursprung, Christopher Banthin, Douglas E. Levy

**Affiliations:** 1Health Policy Research Center, Mongan Institute, Massachusetts General Hospital, Boston, MA 02114, USA; 2Tobacco Research and Treatment Center, Massachusetts General Hospital, Boston, MA 02114, USA; 3Harvard Medical School, Boston, MA 02115, USA; 4Biostatistics Center, Massachusetts General Hospital, Boston, MA 02114, USA; 5Division of General Academic Pediatrics, Massachusetts General for Children, Boston, MA 02114, USA; 6Division of General Internal Medicine, Department of Medicine, Massachusetts General Hospital, Boston, MA 02114, USA; 7School of Public Health, University of Minnesota, Minneapolis, MN 55455, USA; 8Office of Statistics and Evaluation, Bureau of Community Health and Prevention, Massachusetts Department of Public Health, Boston, MA 02108, USA; 9Massachusetts Tobacco Cessation and Prevention Program, Bureau of Community Health and Prevention, Massachusetts Department of Public Health, Boston, MA 02108, USA; 10Public Health Advocacy Institute, Northeastern University, Boston, MA 02115, USA

**Keywords:** tobacco control, health policy, smoke-free policy, secondhand smoke, low income populations, public housing, implementation

## Abstract

A 2018 rule requiring federally-subsidized public housing authorities (PHAs) in the United States to adopt smoke-free policies (SFPs) has sparked interest in how housing agencies can best implement SFPs. However, to date, there is little quantitative data on the implementation of SFPs in public housing. Massachusetts PHAs were among the pioneers of SFPs in public housing, and many had instituted SFPs voluntarily prior to the federal rule. The aim of this study was to examine the adoption, implementation, and outcomes of SFPs instituted in Massachusetts PHAs prior to 2018 using a survey conducted that year. The survey asked if PHAs had SFPs and, if so, what activities were used to implement them: providing information sessions, offering treatment or referral for smoking cessation, soliciting resident input, training staff, partnering with outside groups, using a toolkit, and/or providing outdoor smoking areas. We used multivariable regression to investigate associations between implementation activities and respondent-reported policy outcomes (resident support, complaints about neighbors’ smoking, and the number of violations reported per year). Of 238 Massachusetts PHAs, 218 (91%) completed the survey and 161 had an SFP prior to 2018. Common implementation activities were offering smoking cessation treatment/referral (89%) and information sessions for residents (85%). Information sessions for residents were associated with higher resident support (adjusted odds ratio [AOR] 4.3; 95%CI 1.2–15.3). Training staff (AOR 6.3, 95%CI 1.2–31.8) and engaging in ≥5 implementation activities (AOR 4.1, 95%CI 1.2–14.1) were associated with fewer smoking-related complaints. Utilization of multiple implementation activities, especially ones that informed residents and trained PHA staff, was associated with more favorable policy outcomes. We identified five groups of PHAs that shared distinct patterns of SFP implementation activities. Our findings, documenting implementation activities and their associations with SFP outcomes among the early adopters of SPFs in Massachusetts public housing, can help inform best practices for the future implementation of SFPs in multiunit housing.

## 1. Introduction

Tobacco smoke exposure in multiunit housing is a public health concern, particularly in low-income housing [1]. The 2006 Surgeon General’s report stated that there is no risk-free level of exposure to tobacco smoke [2]. Studies have shown high levels of tobacco smoke exposure in public housing and similar residences and have demonstrated evidence that tobacco smoke drifts from unit to unit [3,4,5,6,7]. 

Following earlier calls for public housing authorities (PHAs) to voluntarily adopt smoke-free policies (SFPs), the United States Department of Housing and Urban Development (HUD) proposed, and in 2018 enacted, the national mandatory smoke-free public housing rule [8,9,10]. This policy applies to residences in buildings owned and operated by local housing authorities with funding from HUD. Smoke-free rules prohibit the use of combustible tobacco products anywhere inside public housing buildings, including residents’ apartments. The rules apply to everyone, including residents, guests, and employees. The federal rule does not, however, stipulate how the policy should be implemented.

Despite the enactment of HUD’s SFP, little is known about how PHAs prepare for the initiation of SFPs, nor is there a PHA-specific evidence base to determine which activities should be included in the implementation. An inappropriate or weak implementation may undermine the effectiveness of SFPs [11,12]. Most studies of SFPs in public housing have focused on the perceptions and attitudes of residents, staff, or owners toward the policy rather than how the policies were implemented and how implementation affected outcomes [13]. Although some prior literature suggested strategies and activities to guide SFP implementation and increase compliance, these suggestions were based on qualitative data such as focus group interviews of residents, case studies, or expert consensus [14,15,16,17,18]. The current study improves this evidence base by providing quantitative evidence on implementation activities around SFPs in public housing using survey data obtained in 2018 from Massachusetts PHAs.

PHAs in Massachusetts have been leaders in the adoption of SFPs in public housing. The first comprehensive SFP in a Massachusetts PHA was implemented in 2011 [19]. In 2012, Boston’s PHA was the first in a major U.S. city to put a comprehensive SFP into effect and was an inspiration for the national smoke-free PHA rule [14]. We surveyed all PHAs in Massachusetts in 2018. Our goal was to develop empirical evidence on best practices in order to guide policymakers and landlords in multiunit housing, including state-funded public housing, subsidized or unsubsidized privately managed buildings, and properties outside the United States, in the development and use of optimal SFP implementation activities. We describe the adoption, implementation, and outcomes of SFPs in Massachusetts PHAs, examine associations between implementation activities and policy outcomes (resident support, complaints about neighbors’ smoking, and the number of violations), and characterize typologies of PHAs that adopted similar implementation activities. We hypothesized that engaging in implementation activities would be associated with more favorable policy outcomes and tested these hypotheses using multivariable regression analyses. 

## 2. Materials and Methods

### 2.1. Sample

Survey data were collected from March to August 2018 by researchers at Massachusetts General Hospital. The sampling frame was built from lists provided by HUD and the Massachusetts Department of Housing and Community Development, including all cities and towns in Massachusetts with PHAs funded by either HUD or the state. After removing closed authorities, duplicates, and housing agencies that did not receive federal or state funding, 238 PHAs were included. 

### 2.2. Survey Methodology

PHA Executive Directors were contacted by email, and if there was no response to the email, by telephone. If Executive Directors could not be reached, the study team attempted to contact another employee of the PHA. Respondents were asked if they felt knowledgeable enough about the PHA’s SFP (or lack thereof) to answer the survey, and if not, they were asked to refer study staff to a more knowledgeable colleague. In cases where a respondent was responsible for multiple PHAs, separate surveys were completed for each. Surveys were completed primarily online using a link sent by email. PHAs contacted by phone could complete the survey online or by phone, according to the respondent’s preference. One survey was completed by mail. All data were recorded using REDCap [20]. PHAs with completed surveys were eligible to be entered into a raffle; at the conclusion of the survey period, twenty PHAs were randomly selected to win a $50.00 gift card they could use to support resident activities. The study was deemed exempt by the Mass General Brigham (formerly Partners HealthCare) Institutional Review Board. 

Respondents were told, “throughout the survey, when we mention a ‘smoke-free rule,’ we mean a rule that says that residents are not allowed to smoke in their apartments.” An SFP was deemed present when respondents answered “yes” to “Does your housing authority have a smoke-free rule at one or more of its properties?” We only include PHAs with uniform policies across their properties. We assessed SFPs that were in place prior to January 2018, when survey distribution commenced. In analyses of SFP outcomes, we further restricted the sample to PHAs that had implemented the policy for at least one year to ensure there was an adequate post-policy experience.

### 2.3. Measures

The survey included a range of questions regarding the PHA’s smoking policy (see Appendix A, Survey Questions). Survey questions were developed and refined by a panel of experts in tobacco policy from academia, the Massachusetts Department of Public Health, pilot study respondents working in public housing, and consultants providing technical assistance to housing authorities. 

### 2.4. Implementation Activities

We asked whether PHAs with SFPs employed a range of implementation activities (yes/no) that had been identified in previous studies as factors potentially associated with the implementation of SFPs [14,15,16,17,18] and then refined by a panel of experts, pilot study respondents and consultants. We classified implementation activities into 7 types: (1) PHA hosted information sessions about the policy for residents, (2) PHA offered residents direct cessation treatment (counselling or medications) or referral to providers’ services, (3) residents were given an opportunity to provide input on implementation before the policy was in effect (i.e., advised where smoking is permitted, advised on enforcement processes, provided education or guidance to fellow residents, helped survey other residents about support for the policy, or advised on inclusion/exclusion of e-cigarettes), (4) PHA provided training to staff about the health effects of secondhand smoke, general information about the policy, providing advice to residents on smoking cessation, resources for helping smokers quit, communication and negotiation skills, resident outreach and engagement, procedures for identifying violations, or procedures for responding to violations, (5) PHA sought technical assistance through partnerships with community organizations (e.g., community service agencies, community health centers, public health departments, etc.), (6) PHA utilized toolkits or materials from HUD or other organizations to guide implementation, and (7), PHA provided any outdoor space where residents were allowed to smoke, including common area patios, designated outdoor smoking areas, in cars or parking lots, and outdoors a fixed distance away from building entrances and doorways. 

### 2.5. Policy Outcomes

We solicited information on three policy outcomes self-reported by respondents: resident support for the policy, changes in resident complaints about neighbors’ smoking, and the number of SFP violations. Respondents rated the current level of support of residents for the SFP as non-supportive, mostly unsupportive, somewhat unsupportive, neither unsupportive nor supportive, somewhat supportive, mostly supportive, or completely supportive. If the respondent reported that residents were mostly or completely supportive of the policies, the level of resident support was coded as “high.” Respondents were also asked whether they received fewer, more, or an unchanged number of resident complaints about neighbors’ smoking after the implementation of the policy. Lastly, the number of violations of the SFP was self-reported. A violation of the SFP was defined as residents smoking in a place where smoking is prohibited, including anywhere inside public housing buildings and apartment units. We asked participants how many violations had occurred since the implementation of the SFPs (categories: 0, 1–5, 5–25, 25–50, 50–75, 75–100, and more than 100). 

### 2.6. Analysis

In PHAs that had SFPs in place prior to January 2018, frequencies and percentages were computed for each implementation activity. In the subset of PHAs with SFPs in place for a year or more at the time of the survey, we used multivariable regression to investigate associations between implementation activities and policy outcomes. To ensure model parsimony, a commonly-accepted and widely used stepwise process was employed to narrow the list of implementation activities included in the models [21]. First, bivariate analyses identified associations between each implementation activity and policy outcome at the *p* < 0.25 level. Implementation activities meeting this preliminary threshold were included in multivariable models. For dichotomous outcomes (high resident support, reduction in complaints), we used multivariable logistic regression adjusting for PHA size and years since SFP enactment as covariates. For the number of violations, we used a censored Poisson model to accommodate interval responses on the survey and adjusted for PHA size and years since SFP enactment using an offset. Due to a transcription error, surveys were distributed with overlapping endpoints (e.g., 1–5, 5–25 rather than 1–5, 6–25). We tested the sensitivity of our findings to alternate methods of coding these intervals. For each outcome, we estimated separate multivariable models where the total number of implementation activities was the independent variable of interest, either as a count (range 0–7) or a dichotomous indicator for a high number of activities (≥5 activities). 

Finally, to identify groups of PHAs adopting similar implementation activities, we used latent class analysis [22]. We explored models with 2–5 latent classes and compared Akaike’s Information Criteria (AIC) and the Bayesian Information Criteria (BIC) to select the number of classes with the best fit to the data. We then examined the characteristics of each group to determine how they differed. All analyses were performed using Stata 16.1 (StataCorp, College Station, TX, USA). 

## 3. Results

Of the 238 surveys distributed, 218 (91%) were completed. We excluded 14 PHAs that did not have uniform SFPs across their properties. Of the 204 remaining PHAs, 161 (79%) had adopted an SFP prior to 2018 (Appendix A). Approximately 41% of PHAs with an SFP were federally funded. The average number of total housing units in PHAs with an SFP was 432.6 (1–50 units, 10%; 51–250 units, 57%; 251–500 units, 15%; 501–1000 units, 9%; 1000 or more units, 9%), while the average number of total housing units in PHAs without an SFP was 162.4 (1–50 units, 23%; 51–250 units, 61%; 251–500 units, 12%; 501–1000 units, 2%; 1000 or more units, 2%). Seventy-four percent of the PHAs that had an SFP prior to 2018 were located in urban areas [23], but all PHAs without SFPs were located in urban areas. 

Table 1 presents the frequency with which the seven implementation activities were put in place in the PHAs that adopted an SFP prior to 2018. Specific sub-components of each implementation activity are listed in Appendix A. The most common implementation activities were offering treatment or referral for smoking cessation before the policy went into effect (89%), followed by information sessions for residents (85%) and partnering with outside groups (65%). Sixty-two percent of PHAs provided staff training for the policy implementation, and 61% solicited resident input for the policy enactment (e.g., advice on enforcement processes). Half of the PHAs reported that they allowed smoking in outdoor areas (i.e., designated areas, common patio, parking lot/in a car, or away from building entrances (15–25 ft)), and 45% used a toolkit provided by HUD or another organization to guide implementation of the smoke-free policy. 

We further investigated PHA-reported resident support, change in residents’ complaints about neighbors’ smoking, and the number of violations of the policy among PHAs that had implemented the policy for one year or more (*n* = 137; Table 2). PHAs with complete data were included in the analyses. Most of the PHA respondents (84%) reported a high level of support for the policy. Providing information sessions for residents was associated with reports of high resident support controlling for resident engagement, the size of housing units, and duration of policy implementation (Adjusted odds ratio [AOR] = 4.3, 95%CI = 1.2–15.3). Few (14%) PHAs reported a reduction in complaints from residents about neighbors’ smoking after SFP enactment. Training staff for implementation was associated with higher odds of a reduction in complaints controlling for the provision of information sessions, the existence of external partnerships, PHA size, and years since policy enactment (AOR = 6.3, 95%CI = 1.2–31.8). On average, PHAs reported 4.9 SFP violations per 100 units per year. There were no individual implementation activities significantly associated with reported violations in the multivariable model. We found that PHAs were more likely to report reduced resident complaints about neighbors’ smoking and fewer violations of the SFPs with each additional implementation activity undertaken (reduced complaints: AOR = 1.6, 95%CI 1.1–2.3; violations: adjusted incidence rate ratio [AIRR] = 0.7, 95%CI 0.6–0.9), and when PHAs had employed five or more implementation activities (reduced complaints: AOR = 4.1, 95%CI 1.2–14.1; violations: AIRR = 0.3, 95%CI 0.1 to 0.7). Results of models with violations as the outcome were not sensitive to alternate characterizations of the interval definitions.

The latent class analysis identified five groups of PHAs with distinct patterns of SFP implementation. Figure 1 presents the proportions of PHAs having each implementation activity in place for each group (see Appendix A for detailed values). For example, PHAs in Group 3 had a 68% probability of providing information sessions for residents when implementing the SFP, while PHAs in Group 5 had a 98% probability of providing information sessions. Group 1 engaged in the basic effort, focusing mainly on offering information sessions and help for smoking cessation (*n* = 46, 29%) and rarely on more time- and resource-intensive efforts like resident engagement and staff training. PHAs in Group 2 (*n* = 20, 12%) sought substantially more resident engagement than Group 1. Compared to Group 1, PHAs in Group 3 (*n* = 19, 12%) were more likely to provide training for staff, seek technical assistance from external organizations, and employ the HUD toolkit. PHAs in Group 4 (*n* = 28, 17%) engaged in substantial staff training but were less likely to seek resident engagement and provide outdoor smoking areas. Group 5 PHAs (*n* = 48, 30%) engaged in comprehensive implementation efforts, utilizing almost all the implementation activities. 

Table 3 compares PHA characteristics across the five PHA groups. PHAs in Group 1 tended to include state-funded housing (few federally-funded units) and a greater proportion of family housing. PHAs belonging to Group 2 also had a high proportion of state-funded units but with fewer housing units and a lower proportion of family housing units. PHAs in Group 1 and Group 2 had higher proportions of rural PHAs (33% and 50%, respectively) compared to PHAs in the other groups (17–26%). Group 3 PHAs were characterized by having a high proportion of family housing (95%), federally-funded housing (63%), and 500 or fewer units (95%). PHAs in Groups 4 and Group 5 generally shared similar characteristics (i.e., urban, family housing), but Group 5 PHAs tended to be the largest (36% with ≥500 units) and, on average, had more recent SFP implementers (2.6 years). 

## 4. Discussion

The current study provides a detailed quantitative analysis of the implementation activities performed by PHAs in Massachusetts who were among the early adopters of SFPs in public housing. We found activities related to providing smoking cessation support and information sessions for residents were common while providing outdoor smoking areas and using an implementation toolkit were less common. Housing managers reported that residents tended to be satisfied with SFPs. Furthermore, specific implementation activities, such as strong communication with residents and staff, were associated with positive outcomes (greater policy support and fewer smoking-related complaints).

Comparing, respectively, our findings with results from a national survey of PHAs who were early SFP adopters, we found similar proportions of PHAs provided smoking cessation support (89% vs. 86%) and solicited resident input (61% vs. 66%), though more provided information sessions (85% vs. 71%) and engaged in partnerships with outside organizations (65% vs. <49%) in Massachusetts, and fewer engaged in staff training (62% vs. 94%) [24]. Data from the New York City Housing Authority collected in 2019 indicate that only 36.1% of residents reported receiving invitations or materials (e.g., flyers) for meetings about SFPs [12]. 

We also found that after the SFP’s enactment, residents’ support for the policy was high. Still, few PHAs reported a reduction in complaints about neighbors’ smoking, and the average number of violations was 4.9 per 100 units per year. These findings appear to be consistent with other studies, although caution is warranted in making comparisons due to differences in the samples (residents vs. executive directors) and details with respect to outcome definitions. Both Jiang et al. and Rokicki et al. found that high initial support for SFPs continued after SFPs went into effect, although residents were not satisfied with enforcement [12,13]. Childs et al. showed that the median of violations of the SPFs among the earliest adopters across the US was five per year [24]. 

According to our results, PHAs who provided information sessions to residents about the policy and who trained staff (e.g., about the rationale and effects of the policy, communication/negotiation skills, and process of enforcement) on average reported better policy outcomes (resident support or reduced complaints about smoking, respectively). This finding aligns with existing research indicating that communicating accurate information about SPFs to the residents is important for implementation [12,14,15]. Monitoring resident compliance with the SFP is key to successful implementation, but some housing authorities have found these efforts to be challenging or inadequate [11,13]. Poor compliance monitoring may stem from a lack of staff skills or resources for implementation [25]. Our findings imply that these barriers may be partially reduced by providing proper training for staff. 

Among the seven implementation activities assessed in our study, offering treatment or referral for smoking cessation was the most common (89% of PHAs). Smoking cessation is not required of smokers living in PHAs with smoke-free rules, but residents perceive that providing proactive support for smoking cessation is an important initial step in implementing the SFP [12,14,16,18]. We did not identify significant associations between the provision of smoking cessation supports and our three study outcomes, but ceiling effects may have limited our ability to demonstrate an association given the fact that nearly all PHA’s offered cessation support. Changes in tobacco use itself, as opposed to tobacco use inside public housing buildings, may be an important legacy of SFP implementation and will be an essential component of evaluations measuring the overall health impact of these policies.

Beyond specific activities, we found that more comprehensive efforts (employing greater numbers of implementation activities) were also associated with reports of a reduction in complaints. Indeed this reflects existing consensus on best practices put forth by HUD and others [14,24,26]. While this particular study found clearer associations with SFP outcomes for some activities compared to others, it should not be assumed that the “non-significant” activities are ineffective and, therefore, should not be implemented. 

Findings from the latent class analysis show five different patterns of implementation, which could be related to local resources. For example, the group of PHAs with the most comprehensive set of implementation activities tended to be larger, located in urban locations, with federal funding and somewhat more recent policy adoption. In contrast, the groups with more limited implementation activities (Groups 1 and 2) were relatively more likely to be smaller in size, located in rural settings, with state funding and relatively earlier policy adoption. 

Smaller PHAs have fewer employees, which is likely why staff training was rare in these groups. Indeed, these results are aligned with findings from the earliest adopters in other areas. Childs et al. found that smaller PHAs were less likely to engage in various implementation activities due to limited staff and financial resources [24]. These findings suggest that small PHAs may benefit from a proactive connection with technical assistance for policy implementation. The groups tending to be comprised of larger PHAs were more likely to have solicited outside technical assistance, perhaps due to greater community connectedness and awareness of the resources available to them. Provision of information sessions on SFPs, which was related to reported resident support, varied by PHA grouping but not in a systematic way. While we found that certain implementation activities were associated with improvements in outcomes reported by survey respondents, understanding the potential for those activities to achieve similar results in other PHAs will require careful consideration of the contexts in which those PHAs operate.

Our specific findings demonstrate that the processes by which SFPs are implemented in complex and diverse systems of public housing vary by the characteristics of the properties. More broadly, our findings make clear that when possible, policy analyses should not simply define policies as binary (present/absent) because policy implementation is frequently heterogeneous; oversimplified analyses may mask important details of policy impact.

The current study has certain limitations. First, while the sample size is large for a study of this type, it is small from a statistical perspective (*n* = 161) and may be underpowered to detect some associations. Future research investigating SFP implementation activities using a large national sample of PHAs is warranted. Second, the outcomes of this study (resident support, complaints about neighbors’ smoking, and the number of violations) are self-reported by PHA staff, not by residents or using administrative records. We strove to ensure that our respondents were well-informed with respect to our survey questions, and results remained consistent when we tested for sensitivity to the addition of controls for respondents’ tenure at their PHAs and the degree to which they were personally involved in the planning and preparation for the SFPs. Nevertheless, there remains the possibility of some inaccuracy or bias in participant responses. Third, associations between implementation activities and outcomes should be interpreted with caution due to the possibility of unmeasured confounding factors. Fourth, data were collected from a single state where most PHAs voluntarily implemented SFPs prior to the national policy mandate; the results may not be generalizable to public housing in other states or where the SFP was mandated. Fifth, data were collected before the COVID-19 pandemic; the results do not reflect the societal disruptions that occurred as a result of the pandemic (e.g., an increase in the amount of time people spent at home during the early phase of the pandemic). Lastly, our survey was not conducted contemporaneously with SFP implementation, so it may be subject to recall bias, though we controlled for the time between policy implementation and survey response. Directions for future research include assessments of involuntarily adopted SFPs, the inclusion of broader geographies and other types of housing, and analyses of data obtained from residents and administrative records.

## 5. Conclusions

Experience from PHAs in Massachusetts that were early adopters of SFPs suggests that meeting with residents, training staff, and engaging in more implementation activities prior to the introduction of an SFP are associated with improved resident support and fewer complaints and including these activities in future implementation efforts may lead to greater policy success. The findings also imply that certain PHAs, particularly smaller ones with fewer resources, may benefit from external support and technical assistance for putting in place a comprehensive set of implementation activities.

## Figures and Tables

**Figure 1 ijerph-20-00078-f001:**
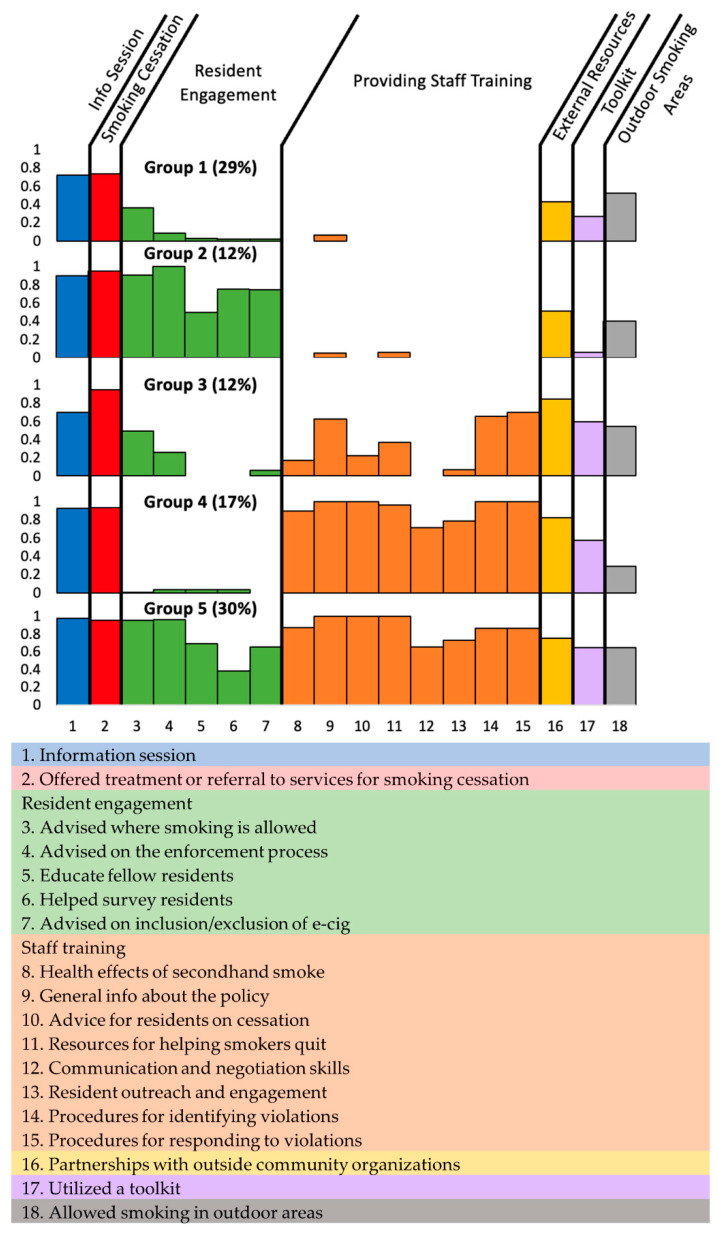
Marginal probabilities of practicing implementation strategies by the five latent classes of public housing from latent class analysis (*n* = 161).

**Table 1 ijerph-20-00078-t001:** Implementation activities in public housing authorities that adopted smoke-free policies prior to January 2018 (*n* = 161).

Implementation Activity	*n* (%)
Offering treatment or referral for help with smoking cessation for residents	143 (89%)
Providing information sessions for residents	129 (85%)
Having a partnership with an outside organization (e.g., community service agencies, local health department, etc.)	105 (65%)
Providing staff training (e.g., health effects of secondhand smoke, a procedure for identifying violations, communication and negotiation skills, etc.)	99 (62%)
Soliciting resident input on the policy implementation	98 (61%)
Providing outdoor smoking areas (i.e., designated areas, common patio, parking lot/in a car, or away from building entrances (15–25 ft))	81 (50%)
Using a toolkit (e.g., from the U.S. Department of Housing and Urban Development or other organizations)	72 (45%)

Note. Ten PHAs (6%) had missing values on providing information sessions for residents.

**Table 2 ijerph-20-00078-t002:** Implementation activities and outcomes among public housing authorities with smoke-free policies for ≥1 year (*n* = 137).

	High Resident Support (Yes: *n* = 103, 84%)		Had a Decrease in Complaints about Neighbors’ Smoking (Yes: *n* = 19, 14%)		Violations per 100 Units a Year ^a^ (Median = 4.9, IQR = 4.7, min = 0, max = 48.7)	
	OR (95%CI)	AOR (95%CI)	OR (95%CI)	AOR (95%CI)	IRR (95%CI)	AIRR (95%CI)
Independent effects of individual activities						
Information session	**3.8** **(1.3, 11.5) ^*^**	**4.3** **(1.2, 15.3) ^*^**	4.2 (0.5, 33.6)	4.2 (0.4, 39.4)	**0.3** **(0.1, 0.6) ^**^**	0.5 (0.2, 1.2)
Smoking cessation aids	1.0 (0.2–4.8)		2.3 (0.3, 19.0)		**0.4** **(0.2, 0.9) ^*^**	1.2 (0.5, 3.0)
Resident engagement	2.3 (0.9–6.2)	2.4 (0.8, 7.3)	1.0 (0.4, 2.7)		**0.3** **(0.1, 0.8) ^*^**	0.6 (0.3, 1.6)
Staff training	0.9 (0.3–2.3)		**6.8** **(1.5, 30.6) ^*^**	**6.3** **(1.2, 31.8) ^*^**	0.7 (0.2, 1.9)	
External partnership	1.5 (0.5–4.0)		2.4 (0.8, 7.7)	1.9 (0.5, 7.1)	**0.2** **(0.1, 0.5) ^**^**	0.6 (0.2, 1.7)
Use of a toolkit	1.0 (0.4–2.8)		0.9 (0.3, 2.3)		**0.3** **(0.1, 0.9) ^*^**	0.6 (0.2, 1.6)
Outdoor smoking	0.7 (0.2–1.8)		1.6 (0.6, 4.3)		0.6 (0.2, 1.6)	
Count of implementation activities						
Number of implementation activities	1.1 (0.9, 1.5)	1.3 (0.9, 1.8)	1.4 (1.0, 2.0)	**1.6** **(1.1, 2.3) ^*^**	n/a	**0.7** **(0.6, 0.9) ^**^**
Used ≥ 5 implementation activities	2.3 (0.6, 9.4)	5.5 (0.8, 38.0)	3.1 (1.0, 9.9)	**4.1** **(1.2, 14.1) ^*^**	n/a	**0.3** **(0.1, 0.7) ^**^**

Note. OR = Odds ratio; AOR = Adjusted odds ratio, models adjusted for all included implementation activities as well as PHA size and years since SFP enactment; CI = Confidence interval; IRR = incidence rate ratio; AIRR = adjusted incidence rate ratio; IQR = interquartile range; min = minimum; max = maximum. Adjusted models controlled for implementation activities that were related to outcomes at *p* < 0.25 in bivariate assessments as well as PHA size (1–50, 51–250, 251–500, 501–1000, >1000) and years since the policy enactment. Boldface indicates statistical significance (* *p* < 0.05, ** *p* < 0.01). There were missing values on resident support (*n* = 15, 11%), complaints about neighbors’ smoking (*n* = 5, 4%), and the number of violations (*n* = 12, 9%). ^a^ Where violations are the outcome, all models are adjusted for PHA size and years since SFP enactment using an offset to standardize comparisons.

**Table 3 ijerph-20-00078-t003:** Characteristics of the five PHA groups as defined by their types of implementation activities (*n* = 161).

	Group 1 (*n* = 46, 29%)	Group 2 (*n* = 20, 12%)	Group 3 (*n* = 19, 12%)	Group 4 (*n* = 28, 17%)	Group 5 (*n* = 48, 30%)	*p* ^a^
	Info Session + Smoking Cessation Help	Group 1 + Efforts for Resident Engagement	Group 1 + Moderate Efforts for Staff Training	Group 1 + Efforts for Staff Training and External Resources	Comprehensive Efforts	
Rural	33%	50%	26%	15%	17%	0.025
Having family housing units on the property (vs. units designated only for elderly/disabled residents)	85%	65%	95%	82%	88%	0.113
Federally funded (vs. state funded)	28%	20%	63%	44%	52%	0.011
Size (total units)						0.069
1–50	17%	15%	11%	7%	2%	
51–250	57%	75%	63%	54%	50%	
250–500	13%	10%	21%	21%	13%	
501–1000	9%	0%	0%	11%	17%	
1000+	4%	0%	5%	7%	19%	
Time since SFP implementation	3.4 years	3.5 years	3.1 years	2.8 years	2.6 years	0.362

^a^ Chi-square test. SFP = smoke-free policy.

## Data Availability

Data will be placed in a publicly accessible archive prior to publication.

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
