# Peer review of "Implementation Activities in Smoke-Free Public Housing: The Massachusetts Experience"

_ijerph, 2022, doi:10.3390/ijerph20010078_

Round 1
Reviewer 1 Report
· What is the main purpose of your study? Define it in the abstract.
· The abstract should be rewritten as to show more the new results and the author’s contribution to the science.
· What does "HUD" mean?
· What is the scientific contribution of your study? The author/s need to do a better job of situating this work in the existing literature and clearly stating the paper’s novel contribution to that literature.
· After your research hypothesis stated in the introduction, describe research methodology applied (just briefly).
· At the end of the introduction, it is necessary to give a brief overview of the structure of the paper.
· Discuss on both theoretical and practical implications of the study.
· In the conclusion of the paper, highlight the key findings of the conducted research.
· It would be interesting to mention the suggestions for future research in terms of new scientific contributions.
· In the end of the paper, References should be written according to the instructions for the authors of the International Journal of Environmental Research and Public Health.
Reviewer 2 Report
This study provides evidence for best practices of SFP implementation in public housing. The methods selected are appropriate to answer research questions. However, as stated in the limitation section, the sample size is small, especially for the latent class analysis in this study (7 indicators & 5 classes).
All data are self-reported by executive directors. Their working experience in SFP and role as a leader may not only influence the validity of the data but also affect the outcomes. The study can be improved if controlling the characteristics of respondents.
Reviewer 3 Report
The authors surveyed public housing authorities (PHAs) in Massachusetts, following a federal mandate that requires smoke-free in federally subsidized public housing. They examined the associations between different activities to implement the smoke-free rule, and policy-relevant outcomes reported by survey respondents. Overall, the paper is well written, and I enjoyed reading it.
The major comments that I have are: the authors should make it very clear at the beginning of the paper that they are examining the implementation of smoke-free policies prior to 2018 in Massachusetts (if that is the case), and furthermore should state clearly the criteria of including a PHA in their sample: does it need to be federally subsidized, and does it need to have smoke-free policy prior to 2018?
Please also see my detailed comments below.
Line 33 – It is unclear what the authors mean by “independently associated” in the abstract.
Line 46 – “HUD” should be fully spelled the first time it was mentioned.
Lines 78 – 83: Are all the PHAs in the sample federally subsidized? Is that one of the eligibility criteria to be included in the sample? Please clarify.
Line 95 – It is worth noting why different survey modes were used, e.g., in what case(s) mail/phone/online surveys were administered.
Line 98 – Regarding the survey questions in Table S3, were survey respondents explicitly asked about the 2018 rule? Since 161 out of 218 PHAs in the sample had smoke-free policies prior to the 2018 rule.
Line 107 – Was the purpose of the survey to examine implementation of smoke-free policies adopted before 2018, to inform the implementation of the 2018 rule? If so, the authors should make that clear in the abstract and introduction.
Line 163 – The authors could also consider using ordered logit model as the alternative specification to test sensitivity, since answers to the survey questions concerning outcome measures are ordinal.
Round 2
Reviewer 1 Report
-
Author Response
It does not appear that there were any further comments from Reviewer 1. We thank the reviewer for her/his review of the manuscript.
Reviewer 3 Report
I have two minor comments:
1. Could you add one sentence in your abstract after introducing the 2018 rule, to clarify that Massachusetts has its own state and local level mandates so that most of the PHAs in your sample had smoke-free policies prior to 2018? It could be right before "The aim of this study...", something like, the fact that most PHAs in MA adopted smoke-free policies in place prior to 2018 makes it a perfect setting to examine the implementation activities.
2. Back to my previous comment about highlighting that this paper examines smoke-free policies prior to 2018, I think the authors added the period "prior to the national SFP" and then deleted it. As a health policy researcher, I think it's important to be as clear as possible what policy and for which time period are you studying.
Author Response
We thank the reviewer for pushing for this important clarification. We have edited the abstract as follows:
"...best implement SFPs. However, to date there is little quantitative data on implementation of SFPs in public housing. Massachusetts PHAs were among the pioneers of SFPs in public housing and many had instituted SFPs voluntarily, prior to the federal rule. The aim of this study was to examine adoption, implementation, and outcomes of SFPs instituted in Massachusetts PHAs prior to 2018 using a survey conducted that year. The survey asked..."